# Extended-Synaptotagmin 1 Enhances Liver Cancer Progression Mediated by the Unconventional Secretion of Cytosolic Proteins

**DOI:** 10.3390/molecules28104033

**Published:** 2023-05-11

**Authors:** Kohji Yamada, Yoshito Hannya, Tsunekazu Oikawa, Ayano Yoshida, Kuniko Katagiri, Saishu Yoshida, Rei Koizumi, Naoko Tago, Yuya Shimoyama, Akira Kawamura, Yuta Mochimaru, Ken Eto, Kiyotsugu Yoshida

**Affiliations:** 1Department of Biochemistry, The Jikei University School of Medicine, 3-25-8 Nishi-Shinbashi, Minato-ku, Tokyo 105-8461, Japan; h30ms-yoshida@jikei.ac.jp (A.Y.); 2Department of Surgery, The Jikei University School of Medicine, 3-25-8 Nishi-Shinbashi, Minato-ku, Tokyo 105-8461, Japan; 3Division of Gastroenterology and Hepatology, Department of Internal Medicine, The Jikei University School of Medicine, 3-25-8 Nishi-Shinbashi, Minato-ku, Tokyo 105-8461, Japan

**Keywords:** unconventional secretion, E-Syt1, hepatocellular carcinoma, tumorigenesis, PKCδ

## Abstract

Extended-synaptotagmin 1 (E-Syt1) is an endoplasmic reticulum membrane protein that is involved in cellular lipid transport. Our previous study identified E-Syt1 as a key factor for the unconventional protein secretion of cytoplasmic proteins in liver cancer, such as protein kinase C delta (PKCδ); however, it is unclear whether E-Syt1 is involved in tumorigenesis. Here, we showed that E-Syt1 contributes to the tumorigenic potential of liver cancer cells. E-Syt1 depletion significantly suppressed the proliferation of liver cancer cell lines. Database analysis revealed that E-Syt1 expression is a prognostic factor for hepatocellular carcinoma (HCC). Immunoblot analysis and cell-based extracellular HiBiT assays showed that E-Syt1 was required for the unconventional secretion of PKCδ in liver cancer cells. Furthermore, deficiency of E-Syt1 suppressed the activation of insulin-like growth factor 1 receptor (IGF1R) and extracellular-signal-related kinase 1/2 (Erk1/2), both of which are signaling pathways mediated by extracellular PKCδ. Three-dimensional sphere formation and xenograft model analysis revealed that E-Syt1 knockout significantly decreased tumorigenesis in liver cancer cells. These results provide evidence that E-Syt1 is critical for oncogenesis and is a therapeutic target for liver cancer.

## 1. Introduction

Protein secretion is fundamental for the development and homeostasis of organisms as well as for the development and progression of cancer. In particular, the secretion mechanisms of various growth factors or several senescence-associated secretory phenotype factors by cancer cells and their surrounding cells are crucial for the formation of the tumor microenvironment [1]. Recently, the extracellular secretion of intracellular proteins without signal peptides, that is, leaderless proteins, termed unconventional secretion, has been reported in inflammation, neurodegenerative diseases, and cancer [2,3]. Many studies have identified several leaderless cytoplasmic proteins in the culture media of cancer cells, including protein kinase C delta (PKCδ), importin α1, and HSP90 [4,5,6,7]. Clinical studies have shown that serum PKCδ levels are significantly higher in patients with hepatocellular carcinoma (HCC) than in normal individuals or those with chronic liver disease (chronic hepatitis and cirrhosis) [8]. Many unconventional secretions, including PKCδ and IL-1β, have been attributed to autophagy-related factors such as ATG5 and ATG7 [3,9]. During secretion, SNAP23, SNAP29, STX3, and STX4 are utilized as soluble *N*-ethylmaleimide-sensitive factor attachment protein receptors (SNAREs) on the plasma membrane (PM) [10,11]. Furthermore, it has been reported to interact with glypican 3 (GPC3) in liver cancer cells, leading to activation of insulin-like growth factor 1 receptor (IGF1R) signaling, which is involved in tumorigenesis [4,12]. However, the relationship between the mechanism of unconventional secretion and hepatocarcinoma tumorigenesis is yet to be elucidated.

Recently, we identified that extended-synaptotagmin 1 (E-Syt1), an integral membrane protein anchored on the endoplasmic reticulum (ER), is involved in the unconventional secretion of PKCδ and importin α1 [9]. E-Syts are a conserved family of tethering factors at the ER-PM contact sites for lipid transport [13]. They can associate with the inner leaflet of the PM mediated by their C-terminal C2 domains; E-Syt1 has five C2 domains, whereas E-Syt2 and 3 have three each [14]. E-Syt2 and 3 localize to the cortical ER, while E-Syt1 localizes throughout the cytoplasmic ER [15,16]. Previous studies reported that E-Syt1 acts as a mediator of cell invasion in non-small cell lung cancer [17]. We also recently identified E-Syt1 as a key factor in the unconventional secretion pathway in liver cancer [9]. Loss of E-Syt1 has been reported to reduce ER localization of PKCδ in liver cancer cells. These findings suggest that E-Syt1 is involved in tumor progression. However, the direct relationship between E-Syt1 and liver cancer remains unclear. In this study, we found that the deletion of E-Syt1 reduced the proliferation and tumorigenic capacity of liver cancer cells. Localization studies revealed that E-Syt1 affected the unconventional secretion of PKCδ and was consequently involved in proliferation signaling triggered by extracellular PKCδ. In addition, xenograft studies have shown that E-Syt1 expression is involved in tumorigenesis.

## 2. Results

### 2.1. E-Syt1 Affects Growth of Liver Cancer Cells

To investigate whether E-Syt1 affects liver cancer growth, we used a HepG2 cell line in which E-Syt1 expression was knocked out [9]. A three-dimensional-tumorsphere formation assay showed that the knockout (KO) of E-Syt1 significantly reduced the number of spheres compared to that in empty vector-transfected control parental cells (control) (Figure 1A,B). We confirmed by RNA interference (RNAi) studies that the knockdown of E-Syt1 significantly reduced attached cell proliferation, albeit only slightly in HepG2 cells (Figure 1C). We further confirmed the anti-growth effects of E-Syt1 deficiency in another liver cancer cell line, HuH7 (Figure 1D–F). These results suggested that E-Syt1 contributes to the growth of liver cancer cells.

### 2.2. E-Syt1 Contributes to PKCδ Secretion in Liver Cancer Cell Lines

Our previous studies have shown that PKCδ is unconventionally secreted in liver cancer cells and in the serum of patients with HCC [4,8,9]. E-Syt1 is also known to play a critical role in PKCδ secretion in HepG2 cells [8]. To confirm the significance of E-Syt1 and cancer-related unconventional secretion in liver cancer cells, we monitored PKCδ secretion in the liver cancer cell lines. Under nutrient starvation conditions, which are similar to most unconventional secretions in other cell types, immunoblot analysis showed that the depletion of E-Syt1 markedly reduced the amount of PKCδ in the culture media of HepG2 and HuH7 cells (Figure 2A,B). Furthermore, we found that the detection of extracellular PKCδ and its suppression by E-Syt1 knockdown were confirmed under physiological conditions (10% FBS) in a cell-based assay using the HiBiT system [9] (Figure 2C), indicating that E-Syt1-mediated unconventional secretion in cancer constitutively occurs.

It is known that E-Syt1 is the main component of the ER-PM contact sites, which functions in a Ca^2+^-dependent manner [16]. Given that several protein secretions (such as neural factors) are affected by intracellular Ca^2+^, we examined whether PKCδ secretion is Ca^2+^-dependent using BaptaAM, a Ca^2+^ chelator. PKCδ secretion was found to be unaffected by BaptaAM treatment (Appendix A), suggesting that PKCδ secretion may be independent of the intracellular Ca^2+^ concentration. It is accepted that, unconventionally, secreted proteins including PKCδ are incorporated into the SEC22B organelle via an autophagy mechanism and then fused with SNAREs, such as STX3, of the PM prior to the secretion [3,9,18]. Indeed, an RNAi study of SEC22B and STX3 showed a remarkable suppression of PKCδ secretion (Figure 2C and Appendix A). Furthermore, a proximity ligation assay (PLA) showed that the knockout of E-Syt1 apparently reduced the interactions between SEC22B and STX3 (Figure 2D), indicating that E-Syt1 appears to be upstream of the unconventional secretion of PKCδ. Taken together, these results indicate that PKCδ undergoes unconventional secretion via interactions with E-Syt1.

It is well known that E-Syt1 localizes to ER in the cell in a dispersed manner [14,15]. We observed colocalization of secreted cytosolic proteins (PKCδ and importin α1) in the ER throughout the cell (Figure 3A and Appendix A). To further visualize the interaction between PKCδ and E-Syt1 in the ER, we established a split-GFP system using doxycycline-induced HepG2 cells. We generated GFP_1–10_-V5-E-Syt1 and PKCδ-GFP_11_-HA constructs [19,20] (Figure 3B), and found that a co-transfection study showed observation of reconstituted GFP signals throughout the cytoplasm (Figure 3C). Confocal microscopic analysis confirmed the co-localization of reconstituted GFP with Sec61β (an ER marker) (Figure 3D). These results indicated that PKCδ localizes to E-Syt1 in the ER.

### 2.3. E-Syt1 Is Involved in the Activation of PKCδ-IGF1R Signaling

Next, we investigated the involvement of E-Syt1 in hepatocarcinoma growth by using E-Syt1-deficient liver cancer cells. First, we confirmed that E-Syt1 deficiency is independent of apoptosis in HepG2 cells (Appendix A). Previous studies have also shown that extracellular PKCδ activates IGF1R signaling involved in GPC3-expressing liver cancer cell lines such as HepG2 and HuH7 [4]. Therefore, we examined the involvement of E-Syt1 in extracellular PKCδ signaling. Knockout of E-Syt1 attenuated the activation of IGF1R and extracellular-signal-related kinase 1/2 (Erk1/2) compared to the control in HepG2 cells (Figure 4A). Similar results were obtained from knockdown experiments in another liver cancer cell line, HuH7 (Figure 4B), indicating that E-Syt1 contributes to the activation of IGF1R-Erk1/2 signaling in GPC3-positive liver cancer cells.

### 2.4. E-Syt1 Is Involved in Liver Cancer Tumorigenesis

To investigate the impact of E-Syt1 on tumorigenesis in liver cancer, we subcutaneously inoculated E-Syt1 KO or control HepG2 cells into NOD/SCID mice. Xenograft mouse model analysis showed that lower tumorigenesis was noted in mice bearing E-Syt1 KO HepG2 cells compared to control cells (Figure 5A,B), indicating that E-Syt1-mediated PKCδ secretion contributes to tumorigenesis. Furthermore, we evaluated the prognostic significance of E-Syt1 expression in patients with HCC. A meta-analysis-based validation was performed using the Kaplan–Meier Plotter (http://kmplot.com/analysis/), in which we accessed on 20 December 2022. The levels of E-Syt1 expression correlated with a poorer prognosis in grade I (hazard ratio = 3.54, *p* = 0.0087) or grade I + II HCC (hazard ratio = 2.38, *p* = 0.017) (Figure 5C), indicating that E-Syt1 expression effected on liver cancer. Taken together, these data strongly suggest that E-Syt1 is involved in hepatocarcinogenesis.

## 3. Materials and Methods

Cell culture. Human liver cancer cell lines (HepG2 and HuH7) were obtained from the Japanese Collection of Research Bioresources in 2017. HepG2 Tet-On^®^ Advanced cells were purchased from Takara (Shiga, Japan) in 2018. HepG2 cells were maintained in Dulbecco’s Modified Eagle’s medium (DMEM; Sigma-Aldrich, St. Louis, MI, USA) supplemented with 10% fetal bovine serum (FBS; Sigma-Aldrich). HepG2 Tet-On^®^ Advanced cells were maintained in α-MEM (Nacalai, Kyoto, Japan) supplemented with 0.1 mM NEAA, 500 μg/mL G418, and 10% Tet system approved FBS (Takara). For starvation, each cell line was washed and cultured in Earle’s Balanced Salt Solution (EBSS; Sigma, St. Louis. USA) for the indicated times. The cell lines were routinely monitored for *Mycoplasma* (4A Biotech Co., Union City, CA, USA). The cells used for the experiments were passaged ten times after thawing.

Reagents. The reagents used were BaptaAM (Dojindo, Kumamoto, Japan), doxycycline (Merck, Darmstadt, Germany).

siRNA siRNA knockdown. Knockdown experiments were performed using On-TARGETplus siRNA SMARTpool (Dharmacon GE) for Non-Targeting Pool (UGGUUUACAUGUCGACUAA, UGGUUUACAUGUUGUGUGA, UGGUUUACAUGUUUUCUGA, UGGUUUACAUGUUUUCCUA), E-Syt-1 (GUACUUGGAUUCAUCAGAA, GUACUACAGUGAAGAACGA, CCAAGACUAUUUCGCAAAC, GCCCUGCUAUCCAUCUAUA), SEC22B (AAUAGUGUAUGUCCGAUUC, GCUAAGCAACUCUUUCGAA, CCUAGAAGAUUUGCACUCA, GAAGCACUCUCAGCAUUGG), or STX3 (GAUCAUUGACUCACAGAUU, AAGAAACUCUACAGUAUCA, AGGGUGAGAUGUUAGAUAA, AAACUCGGCUUAACAUUGA). The cells were transfected using Lipofectamine RNAiMAX (Thermo).

Plasmids. pEGFP-C1-EGFP-ESYT1 (plasmid no. 66830), pHRm-NLS-sCas9-GFP_11 ×_ 7-NLS (plasmid no. 70224), and pHRm-GFP_1–10_-VP64-NLS (plasmid no. 70228) were purchased from Addgene (Cambridge, MA, USA). BioID2-HA was cloned from the MCS-BioID2-HA plasmid (Addgene plasmid no.74224), followed by a 13 × GSGSGS (13 × GS) linker attached to the C terminus of PRKCD (PKCδ) to generate PKCδ-13 × GS-BioID2 in the pTRE vector. For split-GFP fusions, regions of PKCδ-13 × GS-GFP_11_-V5 and GFP_1–10_-HA-E-Syt1 were amplified and cloned into pTRE using a DNA Assembly kit (NEB, MA, USA). For C-terminal HiBiT fusions, the sequence HiBiT (5′-GTGAGCGGCTGGCGGCTGTTCAAGAAGATTAGC-3′) was licensed from Promega (Madison, WI, USA). pTRE-PKCδ-13 × GS-HiBiT was established previously [9].

CRISPR/Cas9-mediated knockout. Human E-Syt1 gRNAs were designed using CRISPRdirect accessed on 7 May 2021 (https://crispr.dbcls.jp accessed on 26 April 2023).

E-Syt1 gRNA: 5′-CGGTGCTGACTTCATTCGGG-3′.

The gRNA was cloned into a lentiviral lentiCRISPR v2 vector (Addgene plasmid no. 98290). This construct was confirmed by DNA sequencing. The E-Syt1 knockout cell line was previously established [9].

Cell-based HiBiT assay. Extracellular localized HiBiT-fused proteins were evaluated using a Nano-Glo HiBiT Extracellular Detection System (Promega) according to the manufacturer’s instructions. Briefly, Tet-On HepG2 cells were plated in 96-well plates and treated with doxycycline for the indicated times. Nano-Glo HiBiT Extracellular Detection reagents were added to all wells, and luminescence was measured after 5 min of incubation using an Infinite 200PRO plate reader (Tecan, Zurich, Switzerland). 

Duolink In Situ Proximity Ligation Assay (PLA). Cells were fixed with 4% paraformaldehyde for 15 min and permeabilized with 0.1% TritonX-100 for 5 min. After blocking, the cells were incubated with mouse antibodies against SEC22B (Santa Cruz sc101267; 1:500) and rabbit antibody against STX3 (Abcam ab133750; 1:500). Fluorescence signals were detected using the Duolink in situ PLA probe according to the manufacturer’s instructions and visualized using a Zeiss LSM 880 laser microscope. The signals were quantified and processed using ImageJ software 1.38e (National Institutes of Health).

Flow cytometry. Cells were dissociated using Accutase (Thermo Fisher Scientific, Waltham, MA, USA) and pelleted by centrifugation at 500× *g* for 5 min at 4 °C. The cell suspensions were incubated with annexin V-FITC and pridium iodide (PI) (Nacalai. Kyoto, Japan). The suspensions were incubated for 30 min at 4 °C. Flow cytometric analysis was performed using MACS Quant (Miltenyi Biotec, North Rhine-Westphalia, Germany). At least three independent experiments were performed. The analysis was performed using the FlowJo software v10.

Immunofluorescence analysis. Cultured cells grown on glass cover slides were fixed in 4% paraformaldehyde for 15 min at 25 °C. Cells were then permeabilized with PBS containing 0.1% TritonX-100 and 1% bovine serum albumin (BSA) for 5 min. The cells were washed thrice with PBS before incubation at 37 °C for 1 h with anti-HA mouse mAb (mouse Bioscience; 1:500), anti-importin α1 mouse mAb (BD HPA016858; 1:200), anti-HA rat mAb (Roche 3F10; 1:500), anti-Sec61β rabbit mAb (CST #14648; 1:200), and anti-calnexin rabbit mAb (CST #2679; 1:200). After washing three times with PBS, the cells were incubated at 25 °C for 30 min with the appropriate Alexa Fluor 488-, 594-, and 647-conjugated secondary antibodies and 4′,6-diamidino-2-phenylindole (DAPI) for nuclear detection. The cells were observed by confocal laser fluorescence microscopy (LSM880, Zeiss, Germany) or fluorescence microscopy (BZ-X800; Keyence, Osaka, Japan).

Split-GFP assay. The split GFP system is based on GFP fragments containing β-strands 1–10 (GFP_1–10_) and β-strand 11 of GFP (GFP_11_), reconstituting the complete β-barrel structure of GFP to emit fluorescence in sufficient proximity. Doxycycline-inducing HepG2 cells were transfected with a single construct to verify the expression, localization, and absence of signals in the GFP channel. The images were represented using pseudo-colors suitable for color-blind palettes.

Proliferation assay. The cells were cultured in 96-well plates at a total volume of 100 μL (5 × 10^3^ cells/well). After 48 h of incubation at 37 °C, 2-(2-methoxy-4-nitrophenyl)-3-(4-nitrophenyl)-5-(2,4-disulfophenyl)-2H-tetrazolium (Nacalai Tesque, Inc., Tokyo, Japan) was added to each well. After 30 min of incubation, the water-soluble formazan dye 1-methoxy-5-methylphenazinium, which is formed upon bio-reduction in the presence of an electron carrier, was measured using an Infinite 200PRO plate reader (Tecan) at 450 nm.

Sphere formation. Cells were seeded in ultra-low-attachment 96-well plates (Corning) to generate spheres. After five days of treatment, spherical cell growth was observed and measured using a BZ-X800 fluorescence microscope. Spheroids with diameters larger than 80 or 100 μm were counted in several random fields of observation. 

Immunoblot. Cells were harvested and resuspended in lysis buffer with or without phosphatase inhibitor (10 mM NaF and 1 mM Na_3_VO_4_). The supernatants were isolated by centrifugation and used as the cell lysates. Equal amounts of proteins were subjected to 10% sodium dodecyl sulphate-polyacrylamide gel electrophoresis (SDS-PAGE) and transferred to a polyvinylidene fluoride (PVDF) membrane. The membranes were probed with the following antibodies: E-Syt1 (ATLAS HPA016858; 1:1000), HA (Roche, Basel, Switzerland, 3F10; 1:500), PKCδ (Abcam ab181076; 1:2000), importin α1 (BD Bioscience, NJ, USA, 610486; 1:1000), and GAPDH (Merck MAB374; 1:2000). Signals were detected using enhanced chemiluminescence (ECL, Thermo Fisher Scientific).

Xenograft studies. All animal studies were approved by the Animal Care and Use Committee of the Jikei University School of Medicine and conducted in accordance with the guidelines for animal experimentation established at the Jikei University School of Medicine (Tokyo, Japan). Cells (5 × 10^6^) in 100 mL of Matrigel (BD Bioscience) were implanted subcutaneously in the back flank of NOD/SCID mice (CLEA, Tokyo, Japan). We used isoflurane in 1 to 2% for inhalant anesthetics at the subcutaneous injection. The tumor size was determined by a caliper measurement of the largest (*x*) and smallest (*y*) perpendicular diameters and was calculated according to the formula *V* = *π*/6*xy*^2^. The mice were sacrificed by isoflurane overdose (4 to 5%) for more than an hour at 26 days after injection of the indicated cells. Euthanasia was confirmed using body temperature and heart rate as parameters.

Statistical analysis and reproducibility. Data are represented as mean ± s.d. from the indicated number of replicates. Statistical analysis was performed using the unpaired, two-tailed Student’s or Welch’s *t*-test, or analysis of variance (ANOVA) tests using Prism 8 software (GraphPad Inc. MA, USA), with *n* and *p* value states in the figure legends.

## 4. Discussion and Conclusions

E-Syt1 is an ER transmembrane protein. In this study, we demonstrated the role of E-Syt1 in the tumorigenesis of liver cancer. E-Syt1 induced the unconventional secretion of cytoplasmic proteins such as PKCδ under both physiological and starved culture conditions. This cancer-related unconventional secretion involves growth signals, including IGF1R and Erk1/2, which directly influence the tumorigenic potential. Therefore, we speculated that ER may play an important role in the development and progression of cancer.

Cancer-related unconventional secretion (CUPS) is different from unconventional secretion in immune and neurodegenerative diseases in several critical aspects [21,22,23,24,25]. First, in contrast to traditional stimulus-responsive unconventional secretions, such as inflammation or starvation, CUPS occurs under unstimulated conditions, i.e., 10% FBS. Second, CUPS may contribute to tumorigenesis. Therefore, we speculate that CUPS can be positioned as a subtype of unconventional secretion, owing to its possible involvement in carcinogenesis.

In this study, we successfully visualized the interaction between PKCδ and E-Syt1 in the ER using the split-GFP system. However, the mechanism by which PKCδ is transported from the E-Syt1 interaction in the ER to the SEC22B-positive organelle involved in secretion is unknown. One possibility is that a large transporter may be recruited and formed on the ER surface before entering the ER. Another possibility is the formation of SEC22B-positive sequestered membrane fractions from the ER (e.g., autophagosomes) [26]. For the latter, unconventional secretion responsive to E-Syt1 (e.g., PKCδ and importin α1) has been shown to be autophagy-factor-dependent [9]. Furthermore, it is known that the unconventional secretion of IL-1β in immune cells is also autophagy-factor- and SEC22B-dependent, although the involvement of E-Syt1 is not clear. Further analysis of membrane trafficking is required.

In the present study, the effect of E-Syt1 deficiency on the inhibition of cell proliferation was not significant. Nevertheless, the phosphorylation of IGF1R and ERK1/2 was decreased in E-Syt1-deficient cells compared to control cells during nutrient starvation. Previous reports have shown that unconventional secretion is induced more dramatically during nutrient starvation than under physiological culture conditions [9], which strongly supports the results of this study. As a potential explanation for the significance of unconventional secretion in cancer, when cancer cells are exposed to the harsh environment associated with the onset of carcinogenesis and the mechanisms of conventional protein secretion, including EGF and VEGF, are moderate, autophagy and unconventional secretion are triggered by starvation to enhance intracellular protein secretion, which is beneficial for cell survival and tumorigenesis. PKCδ is found to be higher in the serum of patients with early-stage HCC [8], suggesting that the unconventional secretion of PKCδ occurs at a stage close to hepatocarcinogenesis. Therefore, this study shows that E-Syt1 not only plays a role in the development of hepatocarcinoma but is also likely to become a therapeutic target for liver cancer.

## Figures and Tables

**Figure 1 molecules-28-04033-f001:**
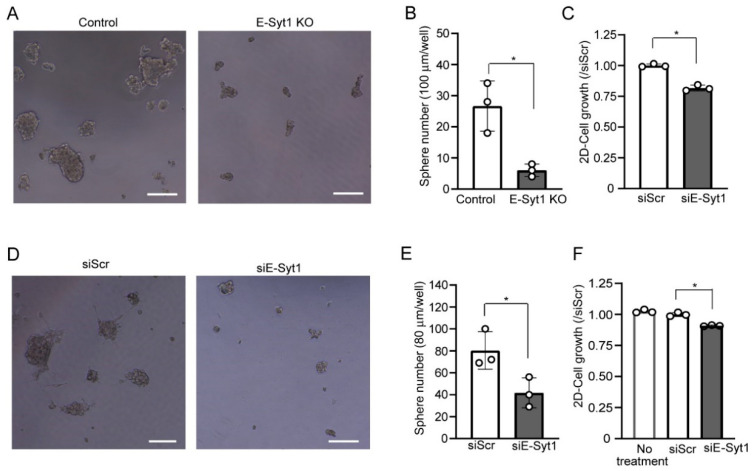
E-Syt1 is involved in the cell proliferation of liver cancer cell lines. (**A**) Images of 3D multicellular spheroids after 5-day culture in control or E-Syt1 KO HepG2 cells showing tumorigenic ability. Scale bars, 100 μm. (**B**) The number of HepG2 3D multicellular spheroids was counted based on the criterion of size > 100 μm. Three independent experiments were performed. Data are shown as the mean ± s.d., * *p* = 0.0407 (Welch’s *t*-test). (**C**) Proliferation of HepG2 cells treated with scrambled (Scr) or E-Syt1 siRNA for 48 h; *n* = 3 independent experiments. Data are shown as the mean ± s.d., * *p* = 0.0011 (Welch’s *t*-test). (**D**) Images of 3D multicellular spheroid after 5-day culture in HuH7 cells treated with scrambled (Scr) or E-Syt1 siRNA to show tumorigenic ability. Scale bars, 100 μm. (**E**) The number of HuH7 3D multicellular spheroids was counted based on the criterion of size >80 µm. Three independent experiments were performed. Data are shown as the mean ± s.d., * *p* = 0.0399 (Welch’s *t*-test). (**F**) Proliferation of HuH7 cells untreated or treated with scrambled (Scr) or E-Syt1 siRNA for 48 h; *n* = 3 independent experiments. Data are shown the mean ± s.d. One-way ANOVA, *p* < 0.0001 with post-test Bonferroni test with pairwise comparison with siScr, * *p* = 0.0001.

**Figure 2 molecules-28-04033-f002:**
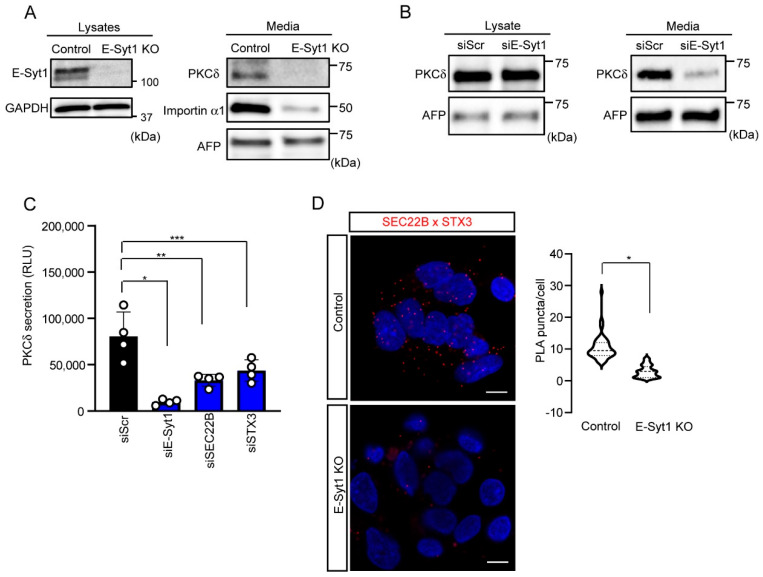
E−Syt1 is required in PKCδ secretion in liver cancer cell lines. (**A**) Immunoblot analysis of lysates or media from starved control or E-Syt1 KO HepG2 cells for 24 h; *n* = 3 independent experiments. Glyceraldehyde 3-phosphate dehydrogenase (GAPDH) and AFP were used as a loading control. (**B**) Immunoblot analysis of lysates and media (24 h) from starved HuH7 cells treated with scrambled (Scr) or E-Syt1 siRNA; *n* = 3 independent experiments. AFP was used as a loading control. (**C**) PKCδ secretion measured by HiBiT extracellular assay in doxycycline-inducible HepG2 cells treated with scrambled (Scr), E-Syt1, SEC22B, or STX3 siRNA for 24 h in 10% FBS condition; *n* = 4 independent experiments. Luminescence was measured after cells were re-cultured in a medium containing 0.5 μg/mL doxycycline for 24 h. Data are shown as the mean ± s.d. One-way ANOVA, *p* = 0.002 with post-test Bonferroni’s test with pairwise comparison with siScr (siE-Syt1, * *p* < 0.0001; siSEC22B, ** *p* = 0.0007; and siSTX3, *** *p* = 0.0041). (**D**) Confocal micrographs to detect the interaction between SEC22B and STX3 (a SNARE on the PM) in control or E-Syt1 KO HepG2 cells. Each cell was fixed, reacted with a combination of mouse anti-SEC22B and rabbit anti-STX3 antibodies (SEC22B × STX3), and subjected to Duolink in situ PLA. Nuclei were stained with 4′,6-diamidino-2-phenylindole (DAPI). Data are shown as the mean ± s.d., * *p* < 0.0001 (two-tailed Student’s *t*-test) (*n* = 49 for control HepG2 cells and *n* = 52 for E-Syt1 KO HepG2 cells). Images represent three independent experiments. Scale bars, 10 μm.

**Figure 3 molecules-28-04033-f003:**
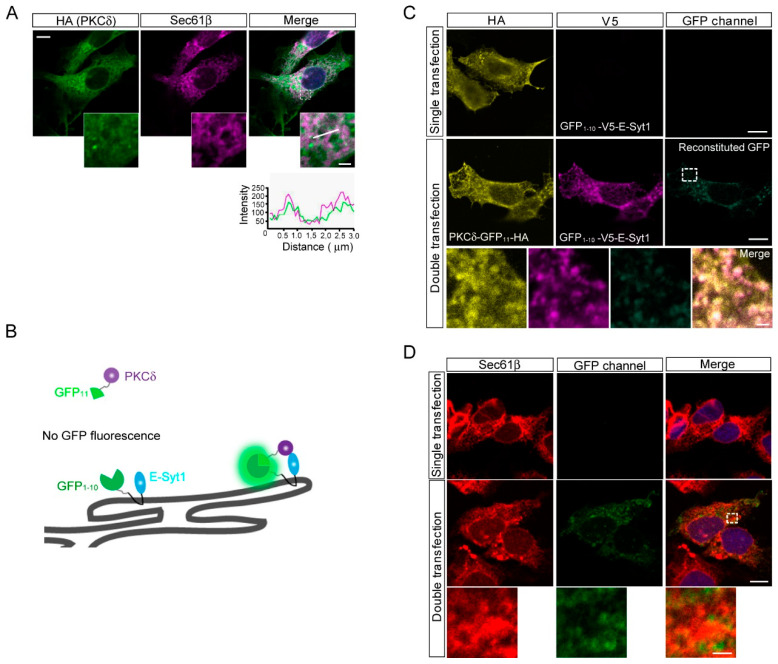
PKCδ interacted with E-Syt1 at the ER. (**A**) Confocal micrographs to detect the co-localization of PKCδ in the ER of HepG2 cells. Sec61β and DAPI were used as ER and nuclear markers, respectively. Scale bar: 10 μm; inset: 2 μm. (**B**) Schematics of the reconstitution of the GFP (split-GFP) system to show co-localization between PKCδ and E-Syt1. (**C**) Split-GFP reconstitution assay between PKCδ-GFP_11_-HA and GFP_1–10_-V5-E-Syt1. Scale bar: 10 μm; inset: 2 μm. (**D**) Split-GFP reconstitution assay between PKCδ-GFP_11_-HA and GFP_1–10_-V5-E-Syt1. Sec61β was used as the ER marker. Scale bar: 10 μm; inset: 2 μm.

**Figure 4 molecules-28-04033-f004:**
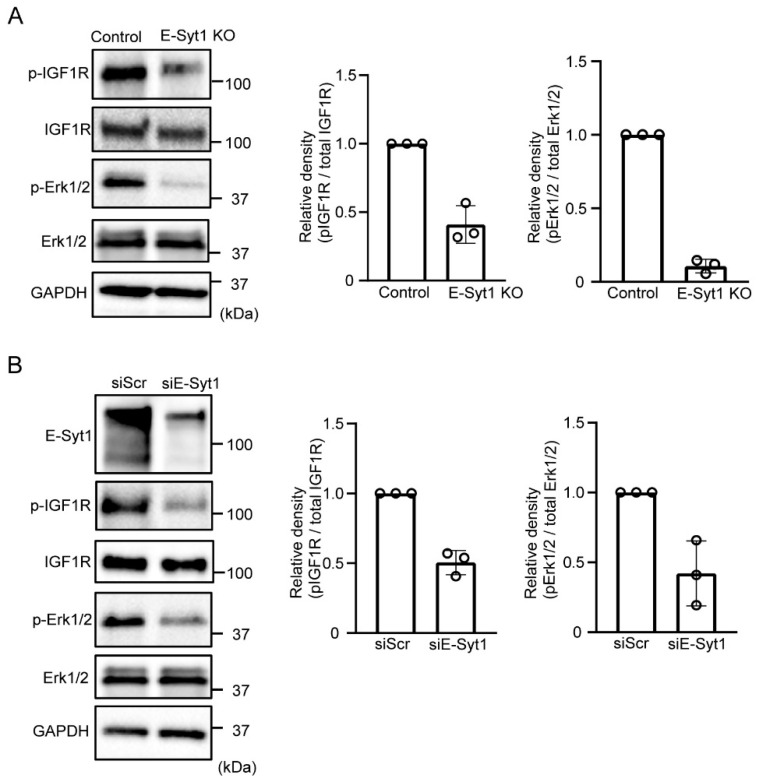
E-Syt1 is required for extracellular PKCδ-IGF1R signaling. (**A**) Immunoblot analysis of phospho-IGF1R (Y1135/1136), phospho-ERK1/2, total IGF1R, total ERK1/2, and Glyceraldehyde 3-phosphate dehydrogenase (GAPDH) (loading control) in the media of starved control or E-Syt1 KO HepG2 cells for 16 h. Three independent experiments were performed. The relative signal density was quantified. (**B**) Immunoblot analysis of phospho-IGF1R (Y1135/1136), phospho-ERK1/2, total IGF1R, total ERK1/2, and GAPDH (loading control) in cell lysates or media of starved HuH7 cells treated with scrambled (Scr) or E-Syt1 siRNA for 16 h. Three independent experiments were performed. Relative signal density was quantified.

**Figure 5 molecules-28-04033-f005:**
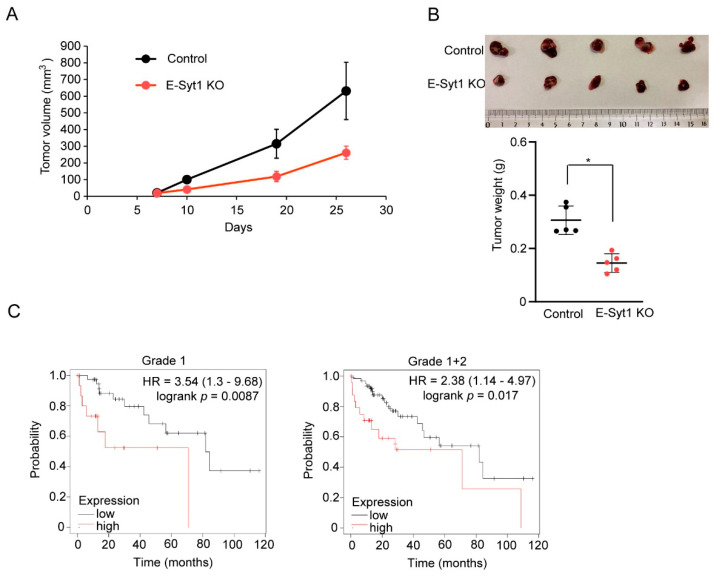
E–Syt1 is involved in the tumor growth of liver cancer cells. (**A**) The control or E-Syt1 KO HepG2 cells were inoculated subcutaneously into NOD/SCID mice (*n* = 5 per group). The tumor size was monitored. (**B**) Microscopic images and tumor weights of control and E-Syt1 KO HepG2 tumors (*n* = 5 per group). Error bars, mean ± s.d., * *p* = 0.0079 (two-tailed Mann–Whitney U test). (**C**) Overall survival was compared between patients with HCC with tumors expressing high levels of *E-Syt1 mRNA* and those expressing low levels. Patients who were negative for hepatitis virus infection were subdivided according to clinical grade (I or I + II). Hazard ratio (HR) and log-rank *p* values are indicated in each panel.

## Data Availability

The original contributions presented in this study are included in the article/Appendix A. Further inquiries can be directed to the corresponding authors.

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
