# Peer review of "Extended-Synaptotagmin 1 Enhances Liver Cancer Progression Mediated by the Unconventional Secretion of Cytosolic Proteins"

_molecules, 2023, doi:10.3390/molecules28104033_

Round 1

Reviewer 1 Report

In the manuscript “Extended-Synaptotagmin 1 Enhances Liver Cancer Progression Mediated by Cancer-Related Unconventional Protein Secretion”, authors aimed to demonstrate the role of E-Syt1 in liver cancer, especially in secretion of PKCδ. The purpose is interesting, timely and has scientific significance. The manuscript is well designed and conducted and draw convincing conclusion. My major concern is that the data are part of a previous study (doi 10.1073/pnas.2202730119), but they are fragmented in a new publication. Item 3.2 “E-Syt1 contributes to PKCδ secretion” should have been included in the previous study and not submitted in a new publication.

My recommendation: Authors should clarify the question raised before the manuscript is published.

Author Response

1, Comment: In the manuscript “Extended-Synaptotagmin 1 Enhances Liver Cancer Progression Mediated by Cancer-Related Unconventional Protein Secretion”, authors aimed to demonstrate the role of E-Syt1 in liver cancer, especially in secretion of PKCδ. The purpose is interesting, timely and has scientific significance. The manuscript is well designed and conducted and draw convincing conclusion. My major concern is that the data are part of a previous study (doi 10.1073/pnas.2202730119), but they are fragmented in a new publication. Item 3.2 “E-Syt1 contributes to PKCδ secretion” should have been included in the previous study and not submitted in a new publication. Extended-Synaptotagmin 1 Enhances Liver Cancer Progression Mediated by Cancer-Related Unconventional Protein Secretion My recommendation: Authors should clarify the question raised before the manuscript is published.

OUR RESPONSE: Thank you for your important remarks.  The previous PNAS paper focused mainly on the mechanism of secretion of cytosolic proteins.  The generality of E-Syt1-mediated secretion in liver cancer cell lines was not clear.  In this respect, the present study differs from previous studies in that it was used with multiple cell lines, including HuH7 cells and newly purchased HepG2 cells, as described in section 3.2, and we consider this to be an important study in that we could confirm the reproducibility of this phenomenon.  The phenomenon itself is novel and it is important to be easily reproducible in the literature.  The point of this research is that E-Syt1, an important factor in the CUPS system, is involved in tumorigenesis, and we consider it possible to apply nucleic acid therapy and genome therapy in the future by studying whether it is a driver gene or not.

In summary, this study is important in cancer research in that it discusses whether E-Syt1 contributes to tumorigenesis, which is different from the previous paper, which focuses on the basic cell biology of secretion mechanisms.  Note that the item 3.2, "E-Syt1 contributes to PKCd secretion," was added as "in liver cancer cell lines," since multiple cell lines were used.

We have added the following phrase to accompany the addition of main text.

Page 5, line 202; “3.2. E-Syt1 contributes PKCd secretion in liver cancer cell lines”

Reviewer 2 Report

Reviewer’s Comments:

The manuscript “Extended-Synaptotagmin 1 Enhances Liver Cancer Progression Mediated by Cancer-Related Unconventional Protein Secretion” is a very interesting work. Extended-synaptotagmin 1 (E-Syt1) is an endoplasmic reticulum membrane protein that is involved in cellular lipid transport. Our previous study identified E-Syt1 as a key factor for the unconventional protein secretion of cytoplasmic proteins in liver cancer, such as protein kinase C delta (PKCd); however, it is unclear whether E-Syt1 is involved in tumorigenesis. Here, we showed that E-Syt1 contributes to the tumorigenic potential of liver cancer cells. E-Syt1 depletion significantly suppressed the proliferation of liver cancer cell lines. Database analysis revealed that E-Syt1 expression is a prognostic factor for hepatocellular carcinoma (HCC). Immunoblot analysis and cell-based extracellular HiBiT assays showed that E-Syt1 was required for the unconventional secretion of PKCd in liver cancer cells. Furthermore, deficiency of E-Syt1 suppressed the activation of insulin-like growth factor 1 receptor (IGF1R) and extracellular signal-related kinase 1/2 (Erk1/2), both of which are signaling pathways mediated by extracellular PKC. The results are consistent with the data and figures presented in the manuscript. While I believe this topic is of great interest to our readers, I think it needs major revision before it is ready for publication. So, I recommend this manuscript for publication with major revisions.

1. In this manuscript, the authors did not explain the importance of Protein Secretionin the introduction part. The authors should explain the importance of Protein Secretion.

2) Title: The title of the manuscript is not impressive. It should be modified or rewritten it.

3) Correct the following statement “Three-dimensional sphere formation and xenograft model analysis revealed that E-Syt1 knockout significantly decreased tumorigenesis in liver cancer cells. These results provide evidence that E-Syt1 is important for unconventional cancer-related secretion and is a therapeutic target for liver cancer”.

4) Keywords: There are so many keywords and reduce them up to 5. So, modify the keywords.

5) Introduction part is not impressive. The references cited are very old. So, Improve it with some latest literature like 10.1002/app.50604, 10.1007/s11837-020-04490-0

6) The authors should explain the following statement with recent references, “It is well accepted that unconventionally secreted proteins including PKCδ are incorporated into the SEC22B organelle via autophagy mechanism and then fused with SNAREs, such as STX3, of the PM prior to the secretion (2, 8, 17)”.

7) Add space between magnitude and unit. For example, in synthesis “21.96g” should be 21.96 g. Make the corrections throughout the manuscript regarding values and units.

8) The author should provide reason about this statement “Similar results were obtained from knockdown experiments in another liver cancer cell line, HuH7 (Figure 4B), indicating that E-Syt1 contributes to the activation of IGF1R-Erk1/2 signaling”.

9) Comparison of the present results with other similar findings in the literature should be discussed in more detail. This is necessary in order to place this work together with other work in the field and to give more credibility to the present results.

10) Conclusion part is very long. Make it brief and improve by adding the results of your studies.

11) There are many grammatic mistakes. Improve the English grammar of the manuscript.

 Minor editing of English language required

Author Response

1, Comment:  In this manuscript, the authors did not explain the importance of Protein Secretionin the introduction part. The authors should explain the importance of Protein Secretion.

OUR RESPONSE: Thank you for your valuable comment.  As Reviewer #2 mentioned, this study shows that this novel protein secretion mechanism is critical for tumorigenesis, and therefore the relationship between the protein secretion and cancer should be described. We would add the statement that protein secretion, especially in the cancer microenvironment, is the keystone of tumorigenesis.  Therefore, the sentence, "In particular, the secretion mechanisms of various growth factors or several senescence-associated secretory phenotype factors by cancer cells and their surrounding cells are crucial for the formation of the cancer microenvironment.", has added in the Introduction.

We have added the following sentence to accompany the addition of main text.

Page 1, line 33; “In particular, the secretion mechanisms of various growth factors or several senescence-associated secretory phenotype factors by cancer cells and their surrounding cells are crucial for the formation of the tumor microenvironment.(1)”

2, Comment: Title: The title of the manuscript is not impressive. It should be modified or rewritten it.

OUR RESPONSE: Thank you for pointing this out.  As Reviewer #2 stated, it is confusing with a list of words, such as Cancer appearing twice, so I decided to change it as follows.

Extended-Synaptotagmin 1 Enhances Liver Cancer Progression Mediated by unconventional secretion of cytosolic proteins

We have modified the original Title as follows.

Revised Title; t Extended-Synaptotagmin 1 Enhances Liver Cancer Progression Mediated by unconventional secretion of cytosolic proteins

3, Comment: Correct the following statement “Three-dimensional sphere formation and xenograft model analysis revealed that E-Syt1 knockout significantly decreased tumorigenesis in liver cancer cells. These results provide evidence that E-Syt1 is important for unconventional cancer-related secretion and is a therapeutic target for liver cancer

OUR RESPONSE: Thanks for the important suggestion.  We have been mixing up our issues. Since we can only say here that E-Syt1 is involved in tumorigenesis, we will also be accurate in our conclusions.  I have corrected it as follows.

 These results provide evidence that E-Syt1 is critical for oncogenesis and is a therapeutic target for liver cancer.

 We have added the following sentence to accompany the addition of main text.

Page 1, line 26; “These results provide evidence that E-Syt1 is critical for oncogenesis and is a therapeutic target for liver cancer.”

4, Comment: Keywords: There are so many keywords and reduce them up to 5. So, modify the keywords.

OUR RESPONSE: Thanks for the comment.  We will delete the sixth split-GFP.

5, Comment: Introduction part is not impressive. The references cited are very old. So, Improve it with some latest literature like 10.1002/app.50604, 10.1007/s11837-020-04490-0

OUR RESPONSE: Thank you for your comment.  This study is a novel mechanism from classical endoplasmic reticulum-associated secretion mechanism and its application to cancer biology, which is why the literature is out of date.  This is also a tribute to the past researchers.  On the other hand, we decided to add the latest literature (Takasugi et al., 2023) to the text with an introduction added regarding protein secretion.  This article shows the latest issues and importance of protein secretion in the tumor microenvironment.  The following is the text to be added.

We have added the following sentence to accompany the addition of main text.

Page 1, line 33; “In particular, the secretion mechanisms of various growth factors or several senescence-associated secretory phenotype factors by cancer cells and their surrounding cells are crucial for the formation of the tumor microenvironment.(1)”

6, Comment: The authors should explain the following statement with recent references, “It is well accepted that unconventionally secreted proteins including PKCδ are incorporated into the SEC22B organelle via autophagy mechanism and then fused with SNAREs, such as STX3, of the PM prior to the secretion (2, 8, 17)”.

OUR RESPONSE: Thank you for your very important comment.  We think ref. 2 and ref. 17 are credible and up-to-date on this mechanism.  At this point in time, most of the work is based on RNAi experiments (ref 2), and the visualization system we are using in this study with PLA staining is the most advanced presentation.  The timing of publication is limited due to the small size of the community and the developing stage of this research field.  On the other hand, the number of researchers interested in this phenomenon has been increasing recently, and the number of references is expected to grow in the future.  Therefore, "well" in the text should be deleted as it may be misleading.

We have depleted the following phase from the main text.

Page 6, line 272; “It is accepted that unconventionally secreted proteins including PKCδ are incorporated into the SEC22B organelle via autophagy mechanism and then fused with SNAREs, such as STX3, of the PM prior to the secretion (3, 9, 18)”

7, Comment: Add space between magnitude and unit. For example, in synthesis “21.96g” should be 21.96 g. Make the corrections throughout the manuscript regarding values and units.

OUR RESPONSE: Thank you for your important comment. We have checked the entire document and found most of the spaces to be correct, but there were some spacing errors, which we have corrected.

8, Comment: The author should provide reason about this statement “Similar results were obtained from knockdown experiments in another liver cancer cell line, HuH7 (Figure 4B), indicating that E-Syt1 contributes to the activation of IGF1R-Erk1/2 signaling

OUR RESPONSE: Thank you for pointing this out.  GPC-positive liver cancer cell lines (HepG2 and Huh7) secrete PKCd, so naturally they should activate IGF1R, and this fact is shown in ref3.  Therefore, we performed an experiment in Fig. 4B to confirm whether the relationship between this result and E-Syt1 deficiency is parallel.  Thus, we added the following sentence to emphasize that the cells are GPC3-positive for clarity.

We have revised the following sentences in the main text.

Page 8, line 310; “Previous studies have also shown that extracellular PKCδ activates IGF1R signaling involved in GPC3-positive some liver cancer cell lines such as HepG2 and HuH7 (4).”

Page 8, line 315; “Similar results were obtained from knockdown experiments in another liver cancer cell line, HuH7 (Figure 4B), indicating that E-Syt1 contributes to the activation of IGF1R-Erk1/2 signaling in GPC3-positive liver cancer cells.”

9, Comment: Comparison of the present results with other similar findings in the literature should be discussed in more detail. This is necessary in order to place this work together with other work in the field and to give more credibility to the present results.”

OUR RESPONSE: Thank you for your valuable suggestion.  The claim of this study suggests that endoplasmic reticulum factor E-Syt1 may be a novel driver gene for liver cancer, which is quite different from other studies.  In this study, we were also able to visualize for the first time that PKCd is actually present in the endoplasmic reticulum.  We were also able to prove inhibition of tumorigenesis using E-Syt1 KO cells.  Although various validations are still needed, we believe that our findings should be able to propose a new therapeutic strategy for cancer treatment.  Therefore, at the beginning of the Discussion, we would like to add the most novel statement, that the endoplasmic reticulum is involved in tumorigenesis, as follows.

We have revised the following sentences in the main text.

Page 9, line 357; “E-Syt1 is an ER transmembrane protein. In this study, we demonstrated the role of E-Syt1 in tumorigenesis of liver cancer. E-Syt1 induced the unconventional secretion of cytoplasmic proteins such as PKCd under both physiological and starved culture conditions. This cancer-related unconventional secretion involves growth signals, including IGF1R and Erk1/2, which directly influence the tumorigenic potential. Therefore, we speculated that ER may play an important role in the development and progression of cancer.”

10, Comment: Conclusion part is very long. Make it brief and improve by adding the results of your studies.

OUR RESPONSE: Thank you.  As Reviewer #2 mentioned, the first and last parts of the sentence are duplicated in meaning.  We will clear it up as follow, except for the second half.

We have deleted the following sentences in the main text.

Page 9, line 408; “This study shows that E-Syt1 deficiency contributes to tumorigenesis, and that E-Syt1 is an important factor for the unconventional secretion of PKCd and is also involved in growth signaling by extracellular PKCd.”

11, Comment: There are many grammatic mistakes. Improve the English grammar of the manuscript.

OUR RESPONSE: Thank you for pointing this out.  We have read several times and corrected them.  We have also received the Editage check and will attach the certificate.

We have revised the following phrases or sentences in the main text.

Page 2, line 61; “glypican 3 (GPC3)”